# Evidence for Biological Age Acceleration and Telomere Shortening in COVID-19 Survivors

**DOI:** 10.3390/ijms22116151

**Published:** 2021-06-07

**Authors:** Alessia Mongelli, Veronica Barbi, Michela Gottardi Zamperla, Sandra Atlante, Luana Forleo, Marialisa Nesta, Massimo Massetti, Alfredo Pontecorvi, Simona Nanni, Antonella Farsetti, Oronzo Catalano, Maurizio Bussotti, Laura Adelaide Dalla Vecchia, Tiziana Bachetti, Fabio Martelli, Maria Teresa La Rovere, Carlo Gaetano

**Affiliations:** 1Laboratory of Epigenetics, Istituti Clinici Scientifici Maugeri IRCCS, Via Maugeri 10, 27100 Pavia, Italy; alessia.mongelli@icsmaugeri.it (A.M.); veronica.barbi@icsmaugeri.it (V.B.); michela.gottardizamperla@icsmaugeri.it (M.G.Z.); sandra.atlante@icsmaugeri.it (S.A.); luana.forleo01@universitadipavia.it (L.F.); 2Foundation “Policlinico Universitario A. Gemelli IRCCS”, Department of Translational Medicine & Surgery, Faculty of Medicine, and Department of Cardiovascular Science, Catholic University of the Sacred Heart, 00168 Rome, Italy; marialisa.nesta@policlinicogemelli.it (M.N.); massimo.massetti@policlinicogemelli.it (M.M.); alfredo.pontecorvi@policlinicogemelli.it (A.P.); simona.nanni@unicatt.it (S.N.); 3Institute for Systems Analysis and Computer Science “A. Ruberti” (IASI), National Research Council (CNR), 00185 Rome, Italy; antonella.farsetti@cnr.it; 4Cardiac Rehabilitation Unit, Istituti Clinici Scientifici Maugeri IRCCS, Via Maugeri 10, 27100 Pavia, Italy; oronzo.catalano@icsmaugeri.it; 5Cardiorespiratory Rehabilitation Department, IRCCS Maugeri Clinical Scientific Institutes, 20097 Milan, Italy; maurizio.bussotti@icsmaugeri.it (M.B.); laura.dallavecchia@icsmaugeri.it (L.A.D.V.); 6Scientific Direction, Istituti Clinici Scientifici Maugeri IRCCS, Via Maugeri 4, 27100 Pavia, Italy; tiziana.bachetti@icsmaugeri.it (T.B.); mariateresa.larovere@icsmaugeri.it (M.T.L.R.); 7Laboratory of Molecular Cardiology, Policlinico San Donato IRCCS, San Donato Milanese, 20097 Milan, Italy; fabio.martelli@grupposandonato.it; 8Department of Cardiology, Istituti Clinici Scientifici Maugeri IRCCS, 27040 Montescano, Italy

**Keywords:** biological age, COVID-19, post-COVID-19, telomeres, epigenetics, DNA methylation, ACE2, DPP-4, DeltaAge

## Abstract

The SARS-CoV-2 infection determines the COVID-19 syndrome characterized, in the worst cases, by severe respiratory distress, pulmonary and cardiac fibrosis, inflammatory cytokine release, and immunosuppression. This condition has led to the death of about 2.15% of the total infected world population so far. Among survivors, the presence of the so-called persistent post-COVID-19 syndrome (PPCS) is a common finding. In COVID-19 survivors, PPCS presents one or more symptoms: fatigue, dyspnea, memory loss, sleep disorders, and difficulty concentrating. In this study, a cohort of 117 COVID-19 survivors (post-COVID-19) and 144 non-infected volunteers (COVID-19-free) was analyzed using pyrosequencing of defined CpG islands previously identified as suitable for biological age determination. The results show a consistent biological age increase in the post-COVID-19 population, determining a DeltaAge acceleration of 10.45 ± 7.29 years (+5.25 years above the range of normality) compared with 3.68 ± 8.17 years for the COVID-19-free population (*p* < 0.0001). A significant telomere shortening parallels this finding in the post-COVID-19 cohort compared with COVID-19-free subjects (*p* < 0.0001). Additionally, ACE2 expression was decreased in post-COVID-19 patients, compared with the COVID-19-free population, while DPP-4 did not change. In light of these observations, we hypothesize that some epigenetic alterations are associated with the post-COVID-19 condition, particularly in younger patients (< 60 years).

## 1. Introduction

SARS-CoV-2-infected people who develop adult respiratory distress syndrome (ARDS) often accumulate an excessive extracellular matrix deposition, causing pulmonary, cardiac, and nervous fibrosis that worsens organ function [1]. Other common features observed in persistent post-COVID-19 syndrome (PPCS) include the increase of circulating troponin T and brain natriuretic peptides (suggesting the presence of myocardium damage with possible activation of a remodeling process) [2]. In addition, the reduction of heart contractility [3], and the alteration of fibrinogen pathways, may lead to an increase in the risk of blood clotting and pulmonary embolism [4,5,6]. Coagulation problems have also been seen in post-COVID-19 survivors and in PPCS patients, to whom anticoagulants are routinely prescribed. Despite the variety and significance of the symptoms reported by numerous COVID-19 survivors with or without PPCS, valuable biomolecular markers to monitor this condition are still lacking.

Upon SARS-CoV-2 infection, the angiotensin-converting enzyme 2 (ACE2) expression level in the vascular system tends to decrease [7]. This enzyme is involved in regulating the renin–angiotensin system (RAS); ACE2 contrasts the activity of the related angiotensin-converting enzyme (ACE) by converting angiotensin II into angiotensin [1,2,3,4,5,6,7,8,9]. A low expression of ACE2 causes an accumulation of angiotensin II, which may exacerbate conditions leading to respiratory distress, hypertension, arrhythmia, cardiac hypertrophy, left ventricular function failure, atherosclerosis, and aortic aneurysms [9,10]. Moreover, ACE2 is negatively correlated with aging; it is relatively abundant in young and healthy people with significantly less risk of CVDs, while a lower quantity is observed in the elderly [11].

Dipeptidyl-peptidase IV (DPP-4) is the receptor of MERS coronavirus (MERS-CoV) and has been reported, in some cases, to function as a coreceptor of SARS-CoV-2 [12]. DPP-4 expression increases on the surface of senescent cells [13], and its transmembrane form can cleave many molecules such as chemokines, neuropeptides, and incretin hormones. DPP-4 inhibitors have been used to treat T2DM, cardiac ischemia, and systolic dysfunction [14,15]. Some evidence indicates that DPP-4 inhibitors might inhibit the entrance of coronavirus into the airways, which suggests an additional therapeutic approach to COVID-19 treatment [16]. Whether the level of ACE2 and DPP-4 in peripheral blood may represent valuable biomarkers to monitor recovery from COVID-19 or the onset of PPCS is unclear.

In humans, telomere shortening is associated in vivo with the aging process and, in vitro, with cellular replicative senescence [17]. Telomeres possess properties that make them suitable as biomarkers in several diseases or conditions, including cancer, CVDs, and aging [18,19]. The inverse correlation between telomere length (TL) and chronological age has been used for age prediction [20]. Interestingly, among individuals infected by Sars-CoV-2, a reduced TL has been associated with the risk of developing more severe symptoms, suggesting that TL at the moment of the infection might influence the clinical outcome [21]. At present, little is known about the telomere dynamics during Sars-CoV-2 infection and in COVID-19 survivors, and whether this parameter might help predict the risk of developing PPCS.

In recent years, several studies aimed to identify biological or molecular markers of aging that correlate with chronological age and could therefore be helpful to estimate biological vs. chronological age [22]. Some of these parameters have been defined based on modifications of the DNA methylome that correlates with chronological age and might be used in age prediction models to define the biological age molecularly: the so-called DNAmAge [23]. Many of these studies focused on healthy or diseased individuals and forensic or public health problems [24,25]. Several methods have been developed to estimate variation in methylation levels in selected DNA CpGs. These approaches apply to determining DNAmAge and have been used to emphasize the difference with chronological age: the so-called DeltaAge. Some methods are based on evaluating many CpGs, explored using a genome-wide array or next-generation sequencing technologies [26]. However, other methods have been developed taking into account the reduced number of CpGs analyzed by pyrosequencing [23,27,28]. All systems are based on DNA methylation values obtained from whole blood samples due to their practicality. These simplified methods have the additional advantage of being rapid and suitable to most laboratory settings without requiring bioinformatics [20,29,30]. Among some of these “reductionist” methods, the algorithm proposed by Bekaert B. et al. performed well for biological age prediction in young and old subjects [20]. This algorithm considers a prediction result correct for individuals aged 60 or higher when the predicted age matches the chronological age within a range of ± 5.2 years [20,31]. Considering that most post-COVID-19 subjects fall within the age group of 50 to 60 years old, or higher, this method was deemed suitable for the present study [20].

A positive DeltaAge is considered an acceleration of the biological blood clock, while a negative DeltaAge indicates a younger blood bioage than the chronological one. This parameter has proven helpful in evaluating the risk of the onset of cardiovascular and neurodegenerative diseases, cancer, and the occurrence of death by all-causes [32].

In infectious diseases, the application of these methods is still limited. However, a DeltaAge acceleration has been observed in people infected by human immunodeficiency virus (HIV), cytomegalovirus, or bacteria such as *Helicobacter pylori* [32]. In post-mortem brain tissue, the DNAmAge of chronically HIV-positive individuals was higher than negative controls. Interestingly, a partial reversion of the accelerated DNAmAge has been observed recently following antiretroviral therapy [33,34]. HIV infection enhances the risk of developing age-related diseases such as neurocognitive disorders [35]. Similarly, in people infected by cytomegalovirus, DNA methylation analyses performed on circulating leucocytes revealed an increased DeltaAge [36]. The long-term consequences of these epigenomic alterations remain to be ascertained.

The present study investigates whether, in COVID-19 survivors, there is a DNAmAge alteration and a DeltaAge acceleration, which, in association with other molecular parameters such as the telomere length and ACE2 expression in peripheral blood, might typify a set of biomarkers valuable in other and future studies exploring the risk of PPCS-associated pathophysiological manifestations.

## 2. Results

### 2.1. Evaluation of DNAmAge and DeltaAge in COVID-19 Survivors

A cohort of 117 COVID-19 survivors came to the attention of our physicians (the clinical features of volunteers are reported in Table 1). Results indicate that the y-axis intercept differs significantly between the COVID-19-free (Figure 1A) and the post-COVID-19 (Figure 1B) populations. The post-COVID-19 group intercepted the y-axis at value 35.22, while the COVID-19-free group intercepted the y-axis at 17.76. This difference determined an increment of DNAmAge of approximately nine years in the post-COVID-19 group, compared with the same group’s chronological age (*p*-value < 0.0001). No difference was appreciable in the controls (Table 2). Accordingly, the vast majority (76.6%) of the post-COVID-19 group had an average DeltaAge acceleration of 10.45 years (Figure 2, red dots). Considering that this method has a tolerance of about ± 5.2 years [20,31], the corrected average accelerated DeltaAge for this group was 5.25. On the other hand, the COVID-19-free volunteers together had a DeltaAge of 3.68, falling well within the range of normality [20] (Figure 2, blue squares). The post-COVID-19/COVID-19-free DeltaAge ratio was 2.84 (Table 2). Interestingly, the DeltaAge distribution within the two groups showed that the COVID-19-free samples were evenly distributed between the normal (39.9%) and the accelerated ranges (48.9%), while the remaining 12.8% had a decelerated biological clock. By contrast, 76.6% of the post-COVID-19 cohort had an accelerated DeltaAge, with only 23.4% falling within the normal or decelerated ranges (Figure 3A). Interestingly, while the COVID-19-free DeltaAge was distributed evenly among the different age groups, the increase of DeltaAge in the post-COVID-19 population was well represented among the younger people (age 56 ± 12.8 years; *p*-value < 0.0001, Figure 3A,B). The older individuals, in both COVID-19-free and COVID-19-survivors groups, did not show signs of DeltaAge acceleration. Interestingly, no differences were noticed between females and males in each age group. This result indicates that the younger survivors might be more sensitive to the SARS-CoV-2-dependent remodeling of the epigenome landscape (Figure 3B).

### 2.2. Telomere Length Quantification

TL shortening has been reported as a risk factor for developing more severe COVID-19 syndrome [21]. We investigated this parameter, which is also associated with the progression of the aging process [19]. Comparing COVID-19-free and post-COVID-19 individuals revealed the presence of a significant shortness of chromosome ends in the COVID-19 survivors’ group (*p*-value < 0.0001; Figure 4A). Specifically, in the COVID-19-free (Figure 4A; blue squares) volunteers, TL was 3.5-fold longer than in the post-COVID-19 group (red dots). The correlation between DeltaAge distribution and TL indicates that post-COVID-19 survivors (Figure 4C; red dots), compared with the COVID-free group (Figure 4B; blue squares), have shorter telomeres (*p*-value < 0.0001) independent of an accelerated DeltaAge, suggesting that these two parameters might be regulated independently. Again, no significant differences emerged between females and males.

### 2.3. Peripheral Blood Expression of ACE2 and DPP-4

In a cell infected by SARS-Cov-2, ACE2 expression decreases, but little is known about the intensity of this biomarker in post-COVID-19 survivors. We evaluated the mRNA level of ACE2 (SARS-CoV and SARS-CoV-2 receptor) and DPP-4 (MERS-CoV receptor). The results are shown in Figure 5A,B. In the post-COVID-19 population, at the time point in which the blood samples were taken, which was more than four weeks from the end of the viral infection (see Table 1), ACE2 expression was significantly reduced (Figure 5A) (*p*-value < 0.0001). The expression level of DPP-4 was unchanged (Figure 5B).

The increased DeltaAge of the post-COVID-19 group correlated well with the lowest ACE2 expression level (Figure 5C; *p* < 0.01). No differences were observed in the distribution of DPP-4 by DeltaAge (Figure 5D). In each group the expression levels of DPP-4 and ACE2 did not change between males and females.

In Table 2 above, we summarize our results and compare COVID-19-free and post-COVID-19 patients.

## 3. Discussion

The global vaccination program against SARS-CoV-2 is actively ongoing, and the incidence of COVID-19 will soon decrease reasonably worldwide. Nevertheless, among the millions of COVID-19 survivors, many will require long-term assistance due to increased post-COVID-19 clinical sequelae defined as PPCS [37,38]. Despite the several manifestations associated with PPCS, there is a lack of potentially valuable molecular biomarkers for the monitoring of PPCS onset and evolution. In this study, we took advantage of the prior indication that biological age, defined as DNAmAge, could be altered in the presence of viral or bacterial infections [33,36,39], and the fact that shorter telomeres are associated with the risk of developing worse COVID-19 symptoms [21]. In this light, we found that a consistently accelerated DeltaAge (5.22 years above the normal range) characterized the post-COVID-19 population, and particularly those chronologically under 60 years (Figure 3A,B). This observation was paralleled by a significant telomere shortening (Figure 4). Although the two parameters seem independent (Figure 4B,C), both alterations coexisted in the post-COVID-19 population. All analyses were performed on blood with a minimally invasive procedure to obtain a source of genetic material exposed to critical environmental changes and associated with the “bona fide” health state of an individual [26,31,40].

However, much remains unknown about the effect of biological age on pulmonary and epithelial health following SARS-CoV-2 infection due to the lack of an appropriate algorithm and the invasive procedure that patients must undergo. The pathophysiology at the basis of these findings remains unclear; however, they may reflect a modified epigenetic environment, particularly evident among the younger COVID-19 survivors (Figure 3). The progression of aging is associated with critical metabolic changes. Some of these changes occur at the level of metabolites regulating the function of essential epigenetic enzymes, such as the decrease in NAD+ levels, the cofactor of sirtuins [41], and the reduction in alpha-ketoglutaric acid [42], the cofactor for all dioxygenases [43]. Although very speculative, it may be that older adults are relatively less sensitive to SARS-CoV-2-dependent epigenetic changes due to changes in their metabolic landscape. Additional experiments are necessary to elucidate this relevant aspect. In light of this consideration, a further question could be whether epigenetic changes might exist antecedent to the first viral contact, persisting or perhaps worsening progressively up to the post-COVID-19 period.

Several epigenetic phenomena have been associated with the SARS-CoV-2 infection [44], including the epigenetic regulation of ACE2 and IL-6. The latter has been associated with the development of worse COVID-19 symptoms due to excessive inflammation [45]. In addition, SARS-CoV-2 has been found to induce changes in DNA methylation, which affect the expression of immune response inhibitory genes that could, in part, contribute to the unfavorable progression of COVID-19 [46]. Finally, it is noteworthy that a recently identified signature made of 44 variably methylated CpGs has been predictive of subjects at risk of developing worse symptoms after SARS-CoV-2 infection [47]. Interestingly, none of these newly identified CpGs overlap with those involved in the DNAmAge prediction used in this [20] or other studies [26]. Hypothetically, it might be possible that distinct signals are regulating the structure of the epigenome regions determining a higher risk of developing a worse COVID-19 syndrome and those associated with DNAmAge prediction.

Even though epigenetics might provide clinically relevant information about COVID-19 [33] progression, no data is currently available regarding the involvement of epigenetic processes in the onset of the post-COVID-19 syndrome or PPCS. Although the post-COVID-19 cohort included in our study was heterogeneous, the range of symptoms observed during the infection varied from mild fever and smelling disturbance to a more severe condition that required assisted ventilation. Our evidence indicates changes in the methylation level of some CpGs associated with biological age calculation. This observation might reflect a more extensive phenomenon underlining unprecedented changes in the epigenome associated with the SARS-CoV-2 infection. A long-term follow-up of patients with an accelerated DeltaAge might help to clarify this critical point.

Telomere length is a marker of aging: progressive telomere shortening is a well-characterized phenomenon observed in older adults and attributed to the so-called telomere attrition. This condition is worsened by the absence of telomerase activity which is physiologically silenced in the early post-natal stage and throughout adulthood [19]. An accelerated TL shortening is a parameter associated with an increased risk of developing cardiovascular diseases and other disorders [48]. In COVID-19, patients bearing shorter telomeres in their peripheral leukocytes have been proposed to be at risk of worse prognoses [49]. In the post-COVID-19 group analyzed here, the average TL was 3.03 ± 2.39 kb, compared with 10.67 ± 11.69 kb in the control group (*p* < 0.0001). As shown in Table 2, the chronological ages of the two cohorts were approximately comparable. Hence, it is unlikely that the aging process was a determinant eliciting the difference. Accordingly, our results suggest that the observed TL shortening could be independent of DeltaAge (Figure 4B,C), indicating that the SARS-CoV-2 infection might directly contribute to telomere erosion in the blood cellular component.

ACE2 is a crucial component of the SARS-CoV-2 infection process. SARS-CoV-2 uses the ACE2 receptor to invade human alveolar epithelial cells and other cells, including cardiac fibroblasts [50]. In infected individuals, ACE2 is often down-regulated due to the infection [7,45]. The enzyme is expressed in several tissues, including alveolar lung cells, gastrointestinal tissue, vascular cells, and the brain; however, it is relatively under-represented in circulating blood cells. In all cases studied, the total relative ACE2 mRNA level in the peripheral blood of non-COVID-19 or post-COVID-19 subjects was significantly lower than that of the MERS-CoV receptor DDP4. However, in the post-COVID-19 group, ACE2 mRNA expression was reduced significantly compared with controls, while DPP-4 demonstrated similar expression levels in both groups. Interestingly, the accelerated DeltaAge, predominant in the younger Post-COVID-19 survivors, significantly correlated with a lower ACE2 mRNA level, suggesting an adverse effect of DNAmAge on ACE2 density in peripheral blood (Figure 5B,C).

The two groups considered in this study were not significantly different in terms of age, sex, and known clinical conditions before SARS-CoV-2 infection, except for a relatively higher incidence of BMI > 30 (15.3% vs. 9%) in the post-COVID-19 population compared with controls, as well as a record of more frequent lung diseases (20.2% vs. 1.6%; see Table 1). The origin of the persistent reduction in ACE2 expression in the post-COVID-19 group remains unsolved, and a longitudinal study should be performed monitoring this parameter.

## 4. Materials and Methods

Upon approval by the Ethical Committee and informed consent signing, peripheral blood was collected in EDTA vacutainers. A group of 144 age- and sex-matched COVID-19-free volunteers with some risk factors partially overlapping with the post-COVID-19 patients were recruited among the hospital workers and non-COVID-19 patients (see Table 1). Genomic DNA was extracted from the whole blood by a robotized station, as described below. After bisulfite conversion and PCR amplification, pyrosequencing was performed. DNAmAge calculations were completed according to Bekaert et al. [20].

The samples were classified into two groups: COVID-19-free (*n* = 144), a heterogeneous group that included healthy, cardiovascular disease-affected, and obstructive sleep apnea-affected patients, and the post-COVID-19 group, which included all of the previous types of patients who had also been infected with SARS-CoV-2 (*n* = 117). The clinical features of both populations are summarized in Table 1.

### 4.1. DNA Extraction from Whole Blood

Blood samples collected in EDTA (200 μL) were used to perform the extraction using the QIAmp DNA Blood Mini Kit (QIAGEN, cat. 55106, Hilden, Germany) associated with the automated system QIACube (QIAGEN, cat. 9002160), according to the manufacturer’s instructions. Subsequently, 2 μL of DNA was quantified with QIAxpert (QIAGEN, cat. 9002340, Hilden, Germany).

### 4.2. Bisulfite Conversion

One microgram of DNA was converted using the EpiTect Fast DNA Bisulfite Conversion Kit (QIAGEN, cat. 59824) associated with the RotorGene 2plex HRM Platform (QIAGEN, cat. 9001560) and the QIACube automated system, following the manufacturer’s instructions. Subsequently, 2 μL of converted DNA was quantified with QIAxpert.

### 4.3. Polymerase Chain Reactions for Pyrosequencing

PCR reaction mixes were performed using the PyroMark PCR Kit (QIAGEN, cat. 978103), following the manufacturer’s instructions. The sequences of primer used are available in the Appendix A.

### 4.4. Pyrosequencing

The amplicons were sequenced in order to check the level of methylation in each CpG site. PyroMark Q24 Advanced Reagents (QIAGEN, cat. 970902) were loaded in the PyroMark Q24 Cartridge (QIAGEN, cat. 979202), following the manufacturer’s instructions, and 5 μL of PCR product was added to the reaction mix containing: Pyromark Binding Buffer (supplied in PyroMark Q24 Advanced Reagents kit), Streptavidin Sepharose High Performance (GE Healthcare, cat. GE17-5113-01), and DNase/RNase-free distilled water. Samples were shaken at room temperature for 15 min at 1400 rpm. Subsequently, the samples underwent the PyroMark Q24 Vacuum Station (QIAGEN, cat. 9001515) procedure, in which the target sequences were purified and put into an annealing buffer containing the sequencing primer (0.375 μM). The sequences of oligos are available in the Appendix A. The plate containing the sequence to analyze and the primer was heated at 80 °C for 5 min. Finally, the PyroMark Q24 Advanced System (QIAGEN, cat. 9001514) was set to analyze the target sequences (available in the Appendix A).

### 4.5. DNAmAge Estimation

Bekaert’s algorithm was applied to estimate the biological age of the population [20] as reported in Daunay et al. [31]:26.44119 − 0.201902 × ASPA − 0.239205 × EDARADD + 0.0063745 × ELOVL22 + 0.6352654 × PDE4C

### 4.6. Telomere Length Quantification

The length of chromosome ends was quantified using a PCR Real-Time of Absolute Human Telomere Length Quantification qPCR Assay Kit (ScienCell, cat. 8918, Carlsbad, CA), following the manufacturer’s instructions.

### 4.7. RNA Extraction

The total RNA was isolated from whole blood using a QIAmp RNA Blood Mini Kit (QIAGEN, cat. 52304) and an automatized extractor QIACube, according to the manufacturer’s instructions. The RNA was quantified with QIAxpert.

### 4.8. cDNA Synthesis and qPCR Real-Time

An Omniscript RT Kit (QIAGEN, cat. 205113) was used to convert total RNA into cDNA according to the manufacturer’s instructions.

Real-time qPCR was performed on the RotorGene 2plex HRM Platform using RT2 SYBR Green ROX FAST Mastermix (QIAGEN, cat. 330620). The sequences of primers are available in the Appendix A. To perform the amplification, the machine settings were:Initial denaturation: 95 °C, 5 min;Denaturation: 95 °C, 15 s;Annealing: 60 °C, 30 s;Elongation: 72 °C, 30 s;Final elongation: 72 °C, 1 min.

Denaturation, annealing, and elongation were repeated 45 times.

### 4.9. Data Analysis

All data were analyzed with GraphPad Prism 8.4.3 and *p*-values were calculated using two-sided T-tests.

## 5. Conclusions

This study has many significant limitations, including the limited number of subjects investigated and the low number of CpGs considered. Although we used a valid forensic method to establish the biological age in the examined groups [20,31], adopting other methods which evaluate a large set of CpGs might be preferable [26,40]. However, the application of such procedures is undermined by the elevated cost and relative complexity and therefore may not be feasible at the laboratory level in many hospitals.

Nevertheless, it was shown here that individuals belonging to a group of COVID-19 survivors exhibited a significant acceleration of their biological age, occurring mainly in the younger individuals. This information was correlated with TL shortening and the expression of ACE2 mRNA. It is too early to extrapolate whether relevant clinical indications may arise from this and other studies assessing the role of epigenetic changes in the COVID-19 syndrome [46,47]. However, a warning might be raised that sequelae of SARS-CoV-2 infection might rely on persistent epigenomic modifications, possibly underlying the presence of a COVID-19 epigenetic memory. The epigenomic landscape of actual post-COVID-19 survivors and prospective COVID-19 survivors from SARS-CoV-2 variants should be considered to gain predictive prognostic insights and monitor more accurately a patient’s response to treatment.

## Figures and Tables

**Figure 1 ijms-22-06151-f001:**
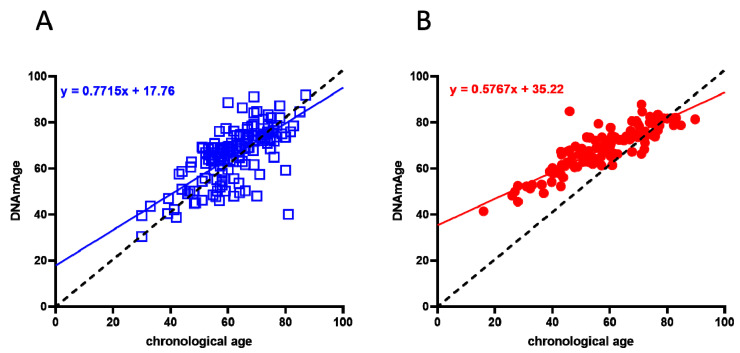
Biological age determination in COVID-19-free (blue squares) and post-COVID-19 (red dots) groups. (**A**) Linear regression of COVID-19-free volunteers’ DNAmAge. (**B**) Linear regression of DNAmAge in the post-COVID-19 subjects. In both graphs, the black dashed line is the bisector and represents the perfect correlation between chronological and biological age. The post-COVID-19 group (right panel) showed a statistically significant DNAmAge acceleration; *p* < 0.0001 (two-sided T-test).

**Figure 2 ijms-22-06151-f002:**
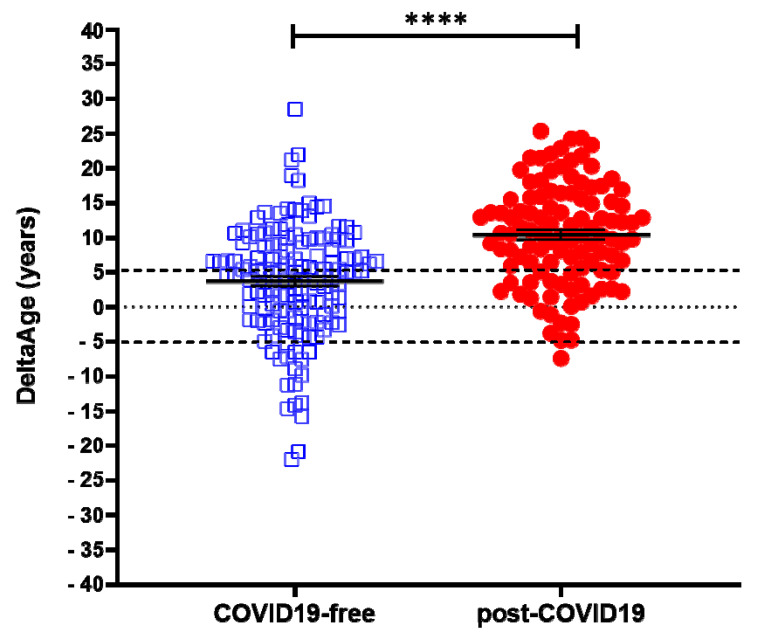
DeltaAge distribution between COVID-free volunteers (left; blue squares) and post-COVID-19 survivors (right; red dots). The black dashed lines indicate the ± 5 years limit of the normal range according to the method. **** *p*-value of < 0.0001 (two-sided T-test).

**Figure 3 ijms-22-06151-f003:**
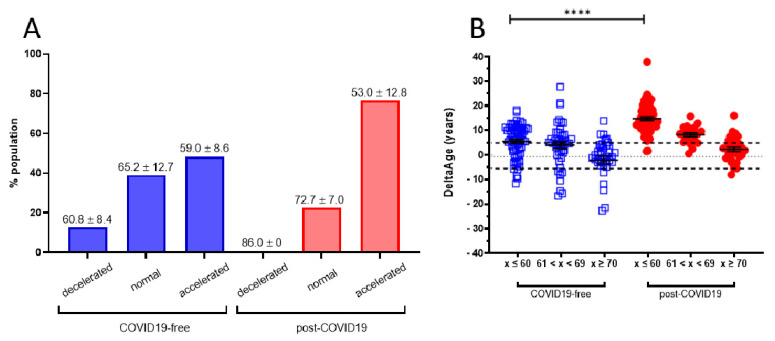
(**A**) DeltaAge range of distribution within each age group. Specifically, in the COVID-19-free cohort (blue bars), 12.8% of the participants were decelerated (mean: −8.7 ± 5); 39.0% fell within the normal range, while 48.2% presented an accelerated DeltaAge (mean: + 5.15 ± 4.34). Interestingly, only a negligible portion (0.9%) of post-COVID-19 patients (red bars) were in the decelerated range. While 22.5% were within the normal range, the vast majority (76.6%) bore an accelerated bioclock (mean: + 8.7 ± 5.79). The average chronological age is reported above each bin. DeltaAge mean values were considered after subtraction of the ± 5.2 normality range distribution. (**B**) The graph shows DeltaAge distribution according to the different chronological age groups. A significance of **** *p* < 0.0001 between the younger COVID-19-free group (< 60) and the corresponding post-COVID-19 patients is shown (two-sided T-test).

**Figure 4 ijms-22-06151-f004:**
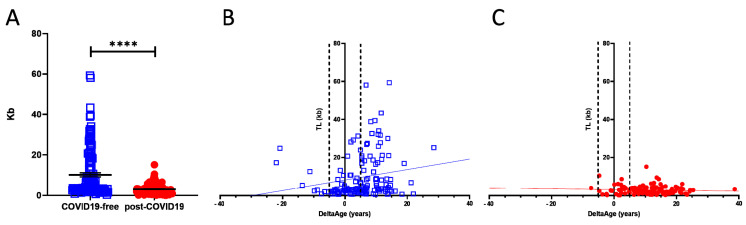
(**A**) Telomere length analysis of COVID-19 survivors (red dots) and COVID-free subjects. The graph shows that the COVID-19-free group has longer chromosome ends compared with the post-COVID-19 group; **** *p*-value < 0.0001 (two-sided T-test). (**B**,**C**) Correlation between DeltaAge and TL in COVID-19-free volunteers ((**B**); blue squares) and post-COVID-19 ((**C**); red dots) patients.

**Figure 5 ijms-22-06151-f005:**
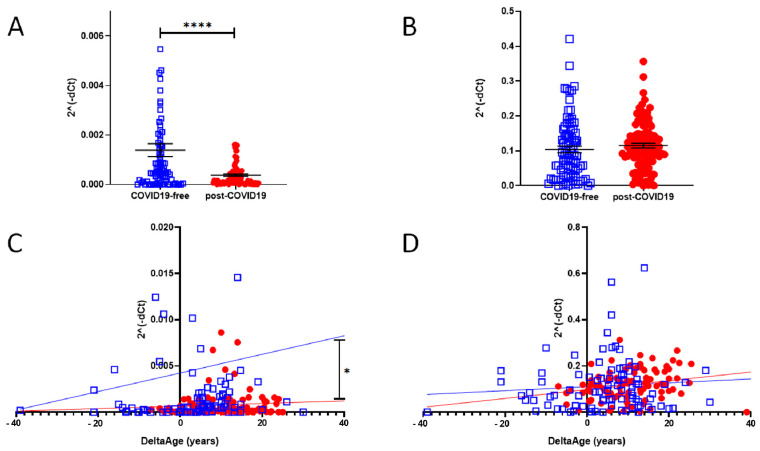
(**A**) qPCR determination of ACE2 expression level in COVID-19-free (blue squares) and post-COVID-19 (red dots) (two-sided T-test: **** *p* < 0.0001). (**B**) mRNA-level determination of DPP-4 in COVID-19-free vs. post-COVID-19, showing no difference between the two groups. (**C**) Correlation between DeltaAge and the relative expression levels of ACE2 mRNA in the peripheral blood of COVID-19-free (blue squares) vs. post-COVID-19 individuals (red dots). * *p* < 0.01 (**D**) Correlation between DeltaAge and the relative expression levels of DPP-4 mRNA in the peripheral blood of COVID-19-free (blue squares) vs. post-COVID-19 individuals (red dots).

**Table 1 ijms-22-06151-t001:** COVID-19 survivors and COVID-19-free volunteers: clinical data.

Clinical Data	COVID-19-Free	Post-COVID-19
Samples (*n*)	144 (Male 66.0%; Female 34.0%)	117 (Male 60.7%; Female 39.3%)
BMI ≥ 30	9.0%	15.3%
Smokers	37.5%	16.9%
Diabetics	11.1%	12.1%
Hypertension	40.3%	36.3%
Clinical history of CVDs	33.3%	27.4%
Antecedent lung involvement	1.6%	20.2%
COVID-19-related complications		
Pneumonia	/	57.3%
Oxygen therapy	/	52.4%
Artificial ventilation	/	35.5%
Length of viral positivity (average) in weeks	/	4.84

**Table 2 ijms-22-06151-t002:** Summary.

	COVID-19-Free	Post-COVID-19	*p*-Value
Samples (*n*)	144 (Male 66.0%; Female 34.0%)	117 (Male 60.7%; Female 39.3%)	
Chronological age (years)	62.48 ± 9.04	58.44 ± 14.66	Ns
Biological age (years)	63.81 ± 13.66	67.18 ± 10.86	Ns
Chronological vs. biological (*p*-value)	Ns	<0.0001	
DeltaAge (years) Ratio	3.68 ± 8.17 1	10.45 ± 7.29 2.84	<0.0001
**DeltaAge distribution**			
Decelerated (%)	12.8	0.9	
Normal (%)	39.0	22.5	
Accelerated (%)	48.2	76.6	
Telomere length (kb)	10.67 ± 11.69	3.03 ± 2.39	<0.0001
ACE2 expression (2^(-dct))	0.001390 ± 0.002298	0.0003801 ± 0.0004463	<0.0001
DPP-4 expression (2^(-dct))	0.1038 ± 0.089	0.1152 ± 0.069	ns

## Data Availability

Data are available from authors upon reasonable request.

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
