# Peer review of "Evidence for Biological Age Acceleration and Telomere Shortening in COVID-19 Survivors"

_ijms, 2021, doi:10.3390/ijms22116151_

Round 1

Reviewer 1 Report

The manuscript of Mongelli  and colleagues reports a study aimed to investigate epigenetic changes associated with the post-COVID19 condition. The authors  have studied  the biological age acceleration and telomere shortening in order to identify a set  of biomarkers involved in post- COVID19 syndrome (PPCS) but however some specific criticisms should be addressed.

  1. The introduction would benefit from shortening to allow a more fluent reading
  2. In the “Materials and methods” section (pag.10, line 323) the clinical features of Samples are not summarized in table 2 but are described in the table 1. The table 2 should be transferred to “Results” section.
  3. I suggest to move the following sentences : Upon approval by the Ethical Committee and informed consent signing, peripheral blood was collected in EDTA vacutainers……… DNAmAge calculations were completed according to Bekaert et al. (“Results” section, pag 4, lines 154-161) in “Materials and methods” section.
  4. It would be appropriate to specify in “Results” section the P-Value and Statistical Significance .
  5. In the section “Results” is not clear if the authors have found sex-specific associations between telomere length and ACE2 expression.

Author Response

To reviewer #1

Thank you for your positive comments, which help to improve the quality of our article.

Q1. The introduction would benefit from shortening to allow a more fluent reading.

A1. As suggested, we edited the text to make it more readable and easier to understand. All your changes have been highlighted in yellow.

Q2. In the "Materials and methods" section (pag.10, line 323) the clinical features of Samples are not summarized in table 2 but are described in the table 1. The table 2 should be transferred to "Results" section.

A2. The typo has been corrected as indicated. Additionally, the correct Table 2 has been moved at the end of the Results section.

Q3. I suggest to move the following sentences : Upon approval by the Ethical Committee and informed consent signing, peripheral blood was collected in EDTA vacutainers……… DNAmAge calculations were completed according to Bekaert et al. ("Results" section, pag 4, lines 154-161) in "Materials and methods" section.

A3. As suggested, the paragraph "Upon approval by the Ethical Committee and informed consent \signing, peripheral blood was collected in EDTA vacutainers………DNAmAge calculations were completed according to Bekaert et al." is now part of the Material and Methods section. However, we did not change the position of Table1 while we added the sentence "the clinical features of volunteers are reported in Table 1" at line126-127.

Q4. It would be appropriate to specify in "Results" section the P-Value and Statistical Significance .

A4.The appropriated p-values have been added for all experiments.

Q5. In the section "Results" is not clear if the authors have found sex-specific associations between telomere length and ACE2 expression.

A5.DeltaAge, telomere length, ACE2, and DPP-4 expressions did not change significantly between males and females as stated in lines 147,180-181, 204-205 of the revised manuscript version.

Reviewer 2 Report

Mongelli et al in this study compared three detections between COVID19 survivors and the normal non-infected people, i.e., the defined CpG island methylation, the telomere length, and the ACE2 receptor expression levels from whole blood cell preparations. The detection methods, the statistical analyses, and the results are straightforward and convincing, though the sample size was not such large. This report adds up a piece of valuable data to COVID19 clinical studies. However, I have some concerns about the samples and the interpretation of how these epigenetics changes resulted from COVID19 infection, which the authors could address.

  1. Were the 117 COVID19 survivors (post-COVID19) infection only (from virus positive becoming negative) without having any disease syndrome (no syndrome infection)? I assume the disease syndrome is different from the PPCS described in the paper. Line 278 “none of the post-COVID19 survivors declared PPCS symptoms but Table 2 indicated the therapy, ventilation, and pneumonia. Then the length and the severity of the disease could be important factors when considering their effect on epigenetics.
  2. Epigenetics changes are driven by environment, in this case, the blood environment. Would the medical intervention via blood injection effect on blood cell DNA methylation? The main target of COVID19 is known to be the epithelia cells.
  3. The whole blood cells are fairly used for biological age estimation of a person at the normal condition. However, in the COVID19 infection, can we say this blood cell detection represent the overall DeltaAge of the patient or only the blood cell type DeltaAge?
  4. The blood cells are known to be produced from bone marrow, and some from lymph nodes, and have a survival time of ~120 days and new generations of blood cells are produced periodically. Is it possible that the epigenetic changes in these blood cells will be reversed to normal after?

Author Response

To reviewer #2

The authors would like to thank the Reviewer for the helpful suggestions.

Q1. Were the 117 COVID19 survivors (post-COVID19) infection only (from virus positive becoming negative) without having any disease syndrome (no syndrome infection)? I assume the disease syndrome is different from the PPCS described in the paper. Line 278 "none of the post-COVID19 survivors declared PPCS symptoms but Table 2 indicated the therapy, ventilation, and pneumonia. Then the length and the severity of the disease could be important factors when considering their effect on epigenetics.

A1. The post-COVID19 cohort of interest in our study is heterogeneous and comprises SARS-CoV-2 subjects that developed symptoms ranging from mild to severe, including some that required artificial ventilation. To evaluate whether disease length and severity might impact the epigenetic features, a more significant number of samples must be collected and possibly during a time course. This study is an initial report, and further studies are ongoing to provide information to this critical question which will be the object of the subsequent publication.

Q2. Epigenetics changes are driven by the environment, in this case, the blood environment. Would the medical intervention via blood injection effect on blood cell DNA methylation? The main target of COVID19 is known to be the epithelia cells.

A2. We want to thank the Reviewer for His/Her insightful comment. Although epithelial cells are targeted by Sars-CoV-2 infection, it is conceivable that the resulting inflammation and immune response might also influence blood cells' epigenetic landscape. Whether pharmacological interventions aimed at controlling the inflammation and the cytokines storm elicited by the infection could also influence the epigenetic profile of blood cells is unknown. In cancer, it seems that the antitumor treatment may indeed introduce changes in the structure of the chromatin. This aspect is still unclear in the presence of anti-Sars-Cov-2 drugs. Although blood is the most straightforward source of genetic material to evaluate the biological clock, recent evidence indicates that specific changes might occur at the organ-specific level. However, epithelial cells are not easily accessible and cannot be evaluated in the present study. A sentence about these interesting points has been added to the Discussion section at lines 229-231

Q3. The whole blood cells are fairly used for biological age estimation of a person at the normal condition. However, in the COVID19 infection, can we say this blood cell detection represent the overall DeltaAge of the patient or only the blood cell type DeltaAge?

A3. Although algorithms exist that measure the multiorgan bioage, most published epigenetic clocks have been optimized for blood. In particular, the forensic method used in this study is based on blood and may be applied to dental material but it is unclear how well it reflects the bioage of other organs. Possibly, the application of this algorithm to other tissues will not correctly reflect the organ bioage. A specific comment has been inserted in the revised version of the manuscript at lines 225-228

Q4. The blood cells are known to be produced from bone marrow, and some from lymph nodes, and have a survival time of ~120 days and new generations of blood cells are produced periodically. Is it possible that the epigenetic changes in these blood cells will be reversed to normal after?

A4. The authors thank the Reviewer for His/Her fascinating question. It is unknown, at present, whether changes in the epigenetic landscape of blood stem cells, once acquired, are permanent or reversible and how long it may take. It is relevant in this light that Bachman et al. reported the accumulation of cytosines modifications in different mouse organs and for a long term. However, little is know about blood stem cells.